# Quantum Dot-Induced Blue Shift of Surface Plasmon Spectroscopy

**DOI:** 10.3390/nano12122076

**Published:** 2022-06-16

**Authors:** Than Thi Nguyen, Vien Thi Tran, Joo Seon Seok, Jun-Ho Lee, Heongkyu Ju

**Affiliations:** 1Department of Physics, Gachon University, Seongnam 13120, Korea; nguyenthan1093@gmail.com (T.T.N.); tranvien04@gmail.com (V.T.T.); wntjs0807@gmail.com (J.S.S.); 2Laser & Opto-Electronics Team, Korea Electronics Technology Institute (KETI), Seongnam 13509, Korea; junholee@keti.re.kr

**Keywords:** resonant coupling, quantum dot, surface plasmon, active medium, Kramers–Kronig relation, optical dispersion

## Abstract

We experimentally demonstrate the spectral blue shift of surface plasmon resonance through the resonant coupling between quantum dots (QDs) and surface plasmons, surprisingly in contrast to the conventionally observed red shift of plasmon spectroscopy. Multimode optical fibers are used for extended resonant coupling of surface plasmons with excited states of QDs adsorbed to the plasmonic metal surface. The long-lived nature of excited QDs permits QD-induced negative change in the local refractive index near the plasmonic metal surface to cause such a blue shift. The analysis utilizes the physical causality-driven optical dispersion relation, the Kramers–Kronig (KK) relation, attempting to understand the abnormal behavior of the QDs-induced index dispersion extracted from blue shift measurement. Properties of QDs’ gain spectrally resonating with plasmons can account for such blue shift, though their absorbance properties never allow the negative index change for the blue shift observed according to the KK relation. We also discuss the limited applicability of the KK relation and possible QDs gain saturation for the experiment–theory disagreement. This work may contribute to the understanding of the photophysical properties critical for plasmonic applications, such as plasmonic local index engineering required in analyte labeling QDs coupled with plasmons for biomedical imaging or assay.

## 1. Introduction

The use of noble metal nanostructures interfaced with local dielectric can transform optical properties of light cone dispersion of a radiation mode into those of non-radiating modes [1,2]. This can generate bound modes of electromagnetic (EM) fields with their dispersion deviating from the light cone, via coherent coupling of conduction electrons with external EM fields, so-called surface plasmons (SPs) that either travel along or are localized at a metal-dielectric interface [3,4,5,6,7]. The EM field energy of surface plasmons is surface confined beyond the diffraction limit due to subwavelength spatial variation of electronic charge density. This enhances the plasmonic electromagnetic field, augmenting the interaction of light with the local dielectric environment such as molecules bound to a metal surface.

For a given metal nanostructure, a local refractive index near the metal surface dictates the optical dispersion of plasmonic fields for which the Debye screening effects dominate the relevant resonance condition [1,2,3,4,5,6,7]. A larger refractive index of local dielectric gives rise to plasmon resonance at longer wavelengths for traveling or localized SPs, called the plasmonic red shift [4,7,8]. Immobilization of adsorbates on metal surfaces or increasing concentration of liquid in contact with the metal surface hence increases the local refractive index, causing the red shift of plasmonic resonance as resulting from light-matter interaction in the surface-confined region.

It is known that the optical dispersion that governs the wavelength dependence of a real part of a refractive index has two different kinds, i.e., the normal and anomalous dispersion according to whether the optical wavelength of interest is far off an optical absorption resonance of a medium or at around it [9]. It is known that, regardless of the dispersion kind, given the composition of a local (absorptive) medium near the metal surface, increasing its concentration always produces the plasmonic red shift in an approximately monotonic manner. The shift magnitude, however, may heavily depend on the wavelength that determines whether the local medium follows the normal or anomalous dispersion [7,10,11].

Particularly, around the absorption resonance (anomalous dispersion), the red shift may occur in a complicated manner such as reported in its oscillatory dependence on wavelength due to the specific energy structure of an absorption-associated electronic transition [10,11]. The Kramers–Kronig (KK) relation can lend itself to the understanding of such non-trivial dependence of the red shift on wavelength. This relation connects real and imaginary parts of the electric susceptibility of a medium in a frequency domain as a result of the physical causality [9,12,13,14,15]. An absorption spectrum characterized by an electronic energy transition structure then determines the non-trivial wavelength dependence of a local refractive index as reported in a resonant coupling between chromophore molecules and localized SPs [10,11].

In this paper, we explore another type of chromophore–plasmon coupling, i.e., quantum dot (QD)–optical fiber SPs systems, focusing on the plasmonic spectral shift. QDs that we choose as chromophores are semiconductor nanocrystals that have characteristics such as a broad absorption spectrum, long-lived excitation, robust photostability, and strong oscillator strength [16,17,18]. The use of optical fiber-based SPs also benefits from fiber propagation-induced extension of QD–plasmon interactions, leading to enhancement of the plasmonic spectral shift.

Meanwhile, the use of QDs that have a broad absorption spectrum above the bandgap energy of a visible wavelength enables one to expect the QD–plasmon coupling to yield the plasmonic red shift at all visible wavelengths according to the KK relation. This is attributed to the positive incremental change in the local refractive index, induced by QDs which have an absorption spectrum extending to an ultraviolet region. This strongly asymmetric absorption spectrum with respect to a given visible wavelength accounts for the positive contribution to a local index change at the wavelength.

We observe, however, the plasmonic blue shifts rather than the red ones surprisingly, and their magnitudes exhibit a non-oscillatory behavior with a wavelength, unlike the oscillatory plasmonic red shifts induced by molecule-localized SPs coupling [10,11]. The experimentally observed blue shifts then employ the analysis of the KK relation, the physical causality-induced principle that can hold valid not only for a passive (absorptive) medium but also for an active (gain) medium [19,20], i.e., QDs excited by a broadband light source. Gain properties of the excited QDs with a long-lived nature of their excited states can contribute to a negative contribution to a local index change, accounting for such blue shifts. A comparison of QD-induced index change between experimental observation and the KK-relation calculation is provided with a discussion on the partial discrepancy between the two. Additional factors related to QDs’ excited states attempt to figure out such disagreements such as possible gain saturation effects and the KK relation’s instabilities in determining the sign of such index change of an active medium, though not violating the physical causality.

## 2. Materials and Methods

We chose a polymer-clad multimode optical fiber with a 200 μm core diameter (JFTLH-Polymicro Technologies, Molex, Lincolnshire, IL, USA) for fabricating a plasmonic optical fiber that can excite fiber-based SPs. First, part of the polymer cladding is eliminated via soldering heat and acetone/methanol to expose the surface of the fiber silica core. Then the bimetallic (consecutively coated layers of 50 nm Ag and 2 nm Au) layers are coated on the exposed core surface [21], as shown in Figure 1A.

A halogen lamp is used as a broadband light source which is made to couple into the optical fiber through an objective lens with 3-axes translational stages. An optical power of less than 7.5 mW is coupled into the fiber. During propagation of optical modes through the bimetal-coated fiber, surface plasmons can be excited at the interface between the outermost metal (Au) and a surrounding medium through an evanescent coupling of light. An optical spectrum of light at the fiber output is measured by a spectrometer (USB-2000, Ocean Optics, Orlando, FL, USA). The measured spectrum that exhibits absorption of light at specific wavelengths is normalized with respect to the spectrum of light through the clad-free fiber with no bimetallic film coated on its core, to highlight spectrally dependent absorption caused by surface plasmons.

A liquid flow cell made of Polydimethylsiloxane [21,22] that encompasses the metal-coated surface enables glycerol solution of various concentrations (from 0 to 20% volume to volume ratio with a 1% step) to be injected for varying a local refractive index near the metal surface from 1.3326 to 1.3634. These refractive indices that correspond to the 21 concentrations are measured by an Abbe refractometer (DR-A1, Atago, Tokyo, Japan), obtaining the linear regression, i.e., n=1.33264+0.00154*C* where *n* is the index and *C* is the solution concentration. The local index variation allows the fiber-based plasmon resonance peak to be spectrally scanned; the plasmonic resonance actually reduces the optical transmittance (T) through the fiber at a certain wavelength band, enabling the absorption spectrum with its peak to be estimated; absorption here defined as 1−T differs from absorbance defined as $\log_{10}\left(1/T\right)$.

After removing the glycerol solution from the cell, the exposed Au surface is then amine-modified with cysteamine (10 mM, incubation of 2 h) for immobilizing carboxylic group-functionalized QDs (CdSe-ZnS, 3 nm/1.5 nm of average diameter, Sigma-Aldrich Co., Rahway, NJ, USA) using the EDC/NHS coupling [23,24]. The core-shell structured QDs exhibit inhomogeneous distributions in size as seen by Gaussian broadening of the fluorescence spectra as shown later (the results and discussion section).

For immobilizing QDs, we immerse the fiber bimetallic surface in the solution with a QD concentration of 0.05 mg/mL (DI water solvent) at 4 °C for 90 min. QD immobilization on the outermost metal (Au) surface can be checked by a fluorescence microscope image as shown in Figure 1B, by means of the fluorescent nature of QDs that emit green light (~560 nm) under a ~470 nm excitation source. After the QDs’ immobilization, subsequent injection of glycerol solution (0–20% concentration) into the flow cell again permits us to scan the plasmonic absorption spectral peak for studying QD–plasmon coupling.

Figure 2 shows the measured spectrum of infrared light absorption in the Fourier transform infrared spectroscopy (PerkinElmer, Spectrum two, Waltham, MA, USA), confirming the presence of carboxyl and amine groups [25,26,27,28] required for QD immobilization on the metal Au surface. In addition, images of immobilized QDs via the atomic force microscopy and the scanning electron microscopy are provided in Appendix A.

## 3. Results and Discussion

### 3.1. QDs Absorbance Spectra and KK Relation

Figure 3 shows the absorbance spectrum of colloidal QDs (concentration of 0.05 mg/mL) measured by a UV-VIS spectrometer (Ultra-3660, Rigol, Portland, OR, USA). The QDs’ absorbance gradually increases for a shorter wavelength in a visible region while drastically increasing for a shorter wavelength in a UV region. This feature, which supports a broad absorbance band due to QDs’ intrinsic energy structures, makes an asymmetric absorbance spectrum with respect to a visible wavelength. The measured spectrum information can lend itself to estimating its index dispersion through the KK relation as follows [13,15]:(1)Im ϵ(ω)=2ωπP∫0∞Re ϵ(ω′)−1ω2−ω′2dω′
(2)Re ϵ(ω)−1=2πP∫0∞Im ϵ(ω′)ω′ω2−ω′2dω′
where ϵ(ω) is the relative permittivity at frequency ω and P denotes the Cauchy principal value. The absorbance spectrum measured allows us to expect that presence of QDs in proximity to the metal surface contributes to positive incremental change in local index (Δn>0) at all visible wavelengths as shown by the KK-relation calculation (see the inset of Figure 3).

### 3.2. Coupling of Fiber Surface Plasmons with QDs

Figure 4A–F show the plasmonic absorption spectra measured without QDs (black curves) and with QDs (red curves) at various glycerol concentrations. Primary peaks in both cases are considered for comparison, though the secondary peaks are not used for comparison as they disappear in cases with QDs. We find that QDs’ adsorption on the plasmonic metal surface causes the blue shift compared to each case without QDs at the same glycerol concentration. In other words, the presence of adsorbed QDs retards the glycerol-caused plasmonic red shift. Thus, QD–plasmon coupling decreases the glycerol-induced red shift. This is manifested in QDs’ effects on a peak wavelength at a given refractive index as shown in Figure 5 where the horizontal axis represents the measured refractive index of a glycerol concentration. This QD-induced “blue shift” is also confirmed by the somewhat left-displaced red curve of an absorption spectrum from the corresponding black one at each glycerol concentration as shown in Figure 4A–F. These surprising results are obviously contradictory to the aforementioned prediction based on the KK relation that adopts the intrinsic (asymmetric) absorbance spectrum of QDs.

### 3.3. QD-Induced Blue Shift of Plasmon Resonance

Let us consider the propagation of a broadband light (a halogen-tungsten lamp used in the setup) along the fiber, which would excite QDs through their interaction with the evanescent field at wavelengths shorter than $\lambda_{G}$. Evanescent field excitation of QDs can occur during fiber mode propagation at wavelengths either close to or away from the plasmon resonance. The long-lived nature of QDs’ excited states, unlike organic dye molecules, then enables the excited QDs to act as a gain medium at around $\lambda_{G}$ due to the population of the excited state carriers being larger than in the ground state as long as a broadband light propagates through the fiber. The QDs’ gain properties, characterized by Im ϵ(ω)<0 in Equation (2), accordingly flip the sign of incremental index changes induced by adsorbed QDs, as opposed to those estimated by the QDs’ intrinsic absorbance spectrum.

In order to obtain the QDs’ gain dispersion information, we measure the photoluminescence (PL) from QDs adsorbed to the bimetallic layer (50 nm Ag, 2 nm Au) deposited on a glass substrate. Glycerol solution is injected into the substrate to vary the local index around QDs. Figure 6A shows the PL spectra at various concentrations. Given a concentration, the Gaussian spectral shape reflects the inhomogenous QDs’ size distribution, its peak slightly shifting from about 564 to 567 nm with varying concentrations from 0 to 20%. We apply the measured PL spectra to the KK relation and calculate the QD-induced index changes as shall be seen in Figure 6B.

As a local medium (glycerol solution) concentration increases, the plasmon absorption peak wavelength increases while the PL spectrum peak wavelength slightly increases. This hence can increase the mismatch between the two peaks, i.e., the resonance mismatch $\Delta \lambda_{R}=\lambda_{\mathrm{peak}}^{\mathrm{SPR}}-\lambda_{\mathrm{peak}}^{\mathrm{PL}}$ used for which QD–SP coupling is considered, to account for the QD-induced spectral shift of plasmon resonance.

Meanwhile, the QD-induced blue shifts observed in Figure 4A–F can be reinterpreted as QD-induced negative index changes. This can be done with an aid of Figure 5 by determining what local index (glycerol concentration) is used to obtain, without QDs, the plasmonic spectral peak wavelength which is the same as the peak wavelength obtained with QDs for a given local index (glycerol concentration). The determination uses the interpolated lines for data (lines connecting solid squares) as given in Figure 5. The QD-induced effective index change can then be found for a given glycerol concentration at which $\Delta \lambda_{R}$ can also be determined experimentally, obtaining the data (solid squares) in Figure 6B. It is unveiled that the QD-induced index changes are all negative over $\Delta \lambda_{R}$ probed with modulating a local medium (glycerol) concentration, which is in accordance with the blue shift effects seen in Figure 4 and Figure 5.

In order to gain an understanding of such negative index changes, we calculate, as mentioned above, the QD-induced index change versus $\Delta \lambda_{R}$ using the KK relation which utilizes the PL spectra, as seen by empty circles in Figure 6B. The calculation with the KK relation provides the negative index changes approximately only at $\Delta \lambda_{R}\;\geq$ 10 nm while producing the positive index changes as $\Delta \lambda_{R}$ tends to be negative.

The fact that the sign of the calculated index changes flips somewhere within the range of $\Delta \lambda_{R}$ probed (as seen by empty circles in Figure 6B) is comprehensible due largely to the quasi-symmetric PL spectra shapes with respect to their spectral centers. It is reminded that, the KK relation with Im ϵ(ω)<0 that uses an ideally symmetric Gaussian PL spectrum, would lead to a QD-induced change in the real part of the local ϵ(ω) that vanishes at the spectral center around which the change flips in sign. This shall result in an anti-symmetric dispersion of the QD-induced index change with respect to the spectral center, similar to an index dispersion derived by the Lorentz–Drude model for resonant interaction of light with bound electron oscillators [9,13,14]. The calculation, however, shows the imperfect features from an asymmetrical shape around $\Delta \lambda_{R}=0$, and this may follow from the fact that KK relation-based calculation uses the measured PL spectra that are subject to practical limitations. This limitation includes the restricted resolution (~±1 nm) in measuring the spectral centers, the insufficient spectral range for Im ϵ(ω), and the imperfect symmetrical shape of the PL spectra.

It is noted that, while the KK relation-based calculation producing the negative index changes at $\Delta \lambda_{R}\;\geq$ 10 nm, it yields positive changes at $\Delta \lambda_{R}<$ 0, not in agreement with the plasmonic blue shift observed experimentally (solid squares), as highlighted by the dashed ellipse in Figure 6B. A number of issues can then be addressed to account for the discrepancy such as the gain saturation of QDs [29] that may occur broadly around $\Delta \lambda_{R}=0$ due to plasmon-supported strong field and an active medium characteristic instabilities in determining the sign of its refractive index based on the KK relation [19,20]. The former is that the condition $\Delta \lambda_{R}\sim 0$ where the strong local fields of plasmonic evanescent fields at photon energies near QDs’ bandgaps can boost the QDs’ gain saturation. This can counteract the gain-induced positive index changes as $\Delta \lambda_{R}$ tends to be negative, resulting in blue shifts particularly as seen by solid squares in Figure 6B. The latter is that, in general, the KK relation-based determination in the refractive index sign for an active medium is not unique [19,20]. This is due to the instability that originates from the fact that not all cases of a medium treated as active satisfy the KK relation, though always meeting the physical causality [19,20]. Such instability may result in the sign flip of the QD-induced index changes as opposed to the KK relation prediction.

It is also worth noting that the use of QDs as the medium that couples with SPs rather than fluorescent dye molecules cause such plasmonic blue shift, due largely to the long-lived characteristics of excited QDs’ states compared to excited dye molecules.

## 4. Conclusions

We demonstrate the QD-induced blue shift in absorption spectroscopy of the fiber-based surface plasmons as a result of QD–plasmon resonant coupling. The negative refractive index changes responsible for such a blue shift can be induced by QDs adsorbed to the plasmonic metal surface. The KK relation is employed to understand the relevant optical dispersion that dictates the QD-induced blue shift of surface plasmons. We find that the spectral structure of the QDs’ absorption cannot account for the blue shift at any visible wavelength but their emission structure (gain characteristics) can in part of the visible wavelength region. The partial qualitative discrepancy between experiment and theory when using the KK relation for QDs as active media may arise from factors such as QDs’ gain saturation assisted by plasmon producing strong fields and KK relation inherent instabilities in determining the sign of an active medium index.

It is noted that optical power-dependent study of the blue shift has been limited by the fact that an increase in fiber incident power beyond a certain threshold may cause photobleaching of QDs due to both enhanced intensity of plasmon evanescent field and its subsequent thermal effects in close proximity that damage QDs.

The use of different sized QDs is expected to induce a different magnitude of blue shift due to the fact that size-dependent electronic energy structure would affect the absorption and emission properties as well as excited state lifetime.

Understanding the surface plasmon–chromophore resonant coupling can be critical for a plasmonic application that relies on local index engineering. The work presented may contribute to understanding particularly the refractive index-dependent photophysical properties of analyte labeling QDs coupled with plasmons.

## Figures and Tables

**Figure 1 nanomaterials-12-02076-f001:**
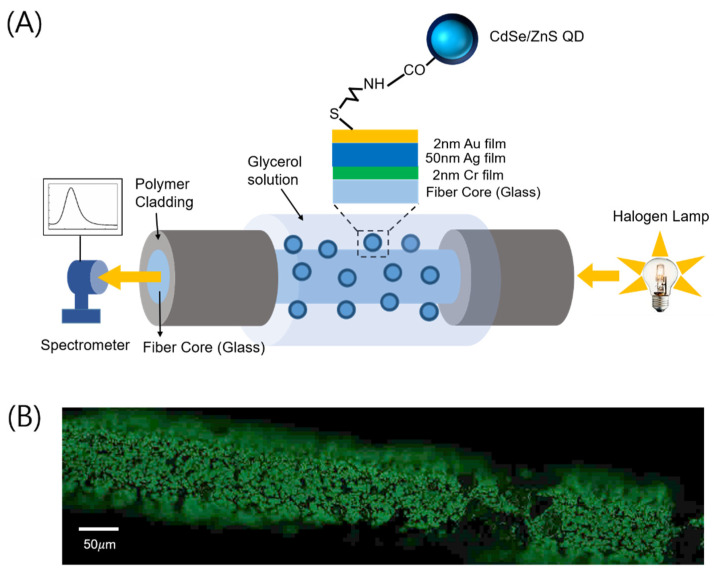
(**A**) Schematic for measuring the plasmonic absorbance spectrum of light through an optical fiber. (**B**) Fluorescence image due to QDs adsorbed on the Au surface of the exposed fiber core under an illumination of 470 nm LED.

**Figure 2 nanomaterials-12-02076-f002:**
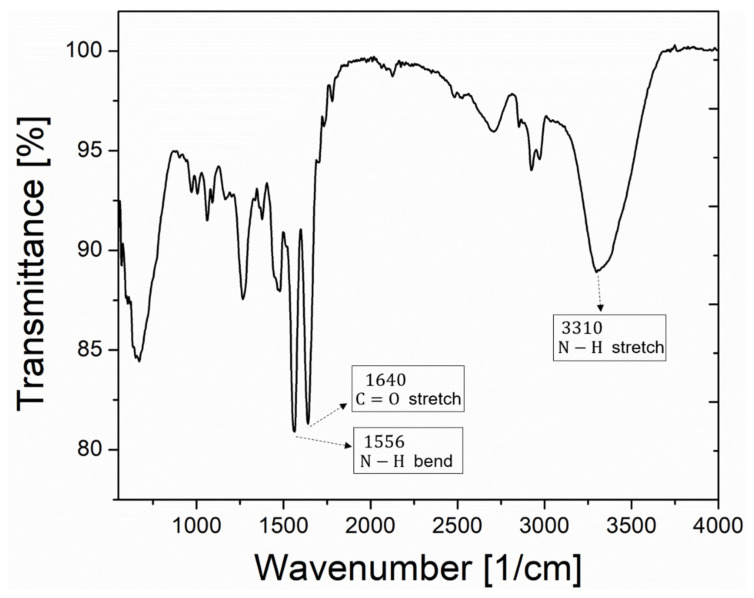
Infrared absorption spectrum for the QDs-immobilized bimetallic film (QDs-2 nm Au-50 nm Ag) on a glass substrate. A number of absorption peaks as indicated by arrows supports the presence of amine-carboxylic bonds [25,26,27,28] required for QD immobilization.

**Figure 3 nanomaterials-12-02076-f003:**
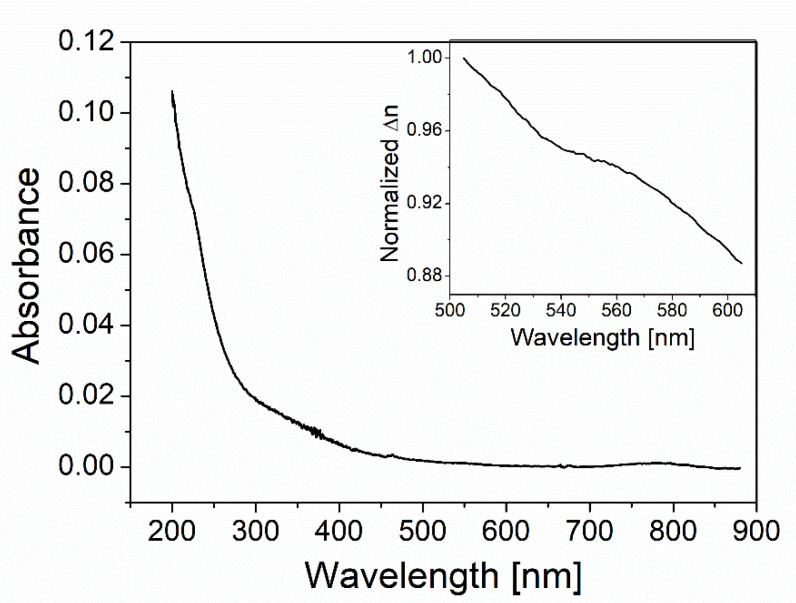
A measured absorbance spectrum of colloidal QDs. The absorbance equals $\log_{10}\left(1/T\right)$ where T is the optical transmittance. The inset shows the normalized value of Δn(>0 ) due to the QDs’ contribution to a local refractive index, being estimated with no scale factor through the KK relation.

**Figure 4 nanomaterials-12-02076-f004:**
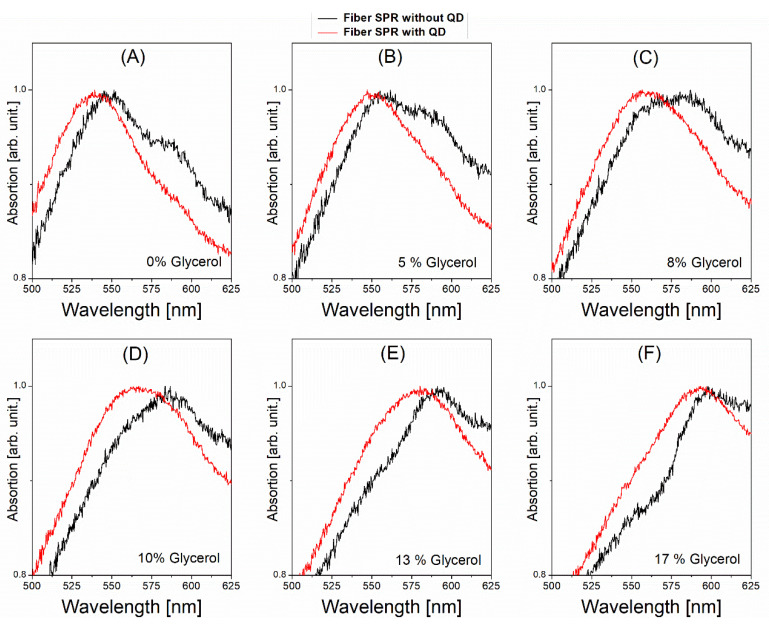
Fiber-based plasmonic absorption spectra at various concentrations of glycerol solution, (**A**) 0% (**B**) 5% (**C**) 8% (**D**) 10% (**E**) 13% (**F**) 17%. Black and red curves represent cases without and with QDs anchored on the outermost metal (Au) surface, respectively. The fiber incident optical power is 3.8 mW.

**Figure 5 nanomaterials-12-02076-f005:**
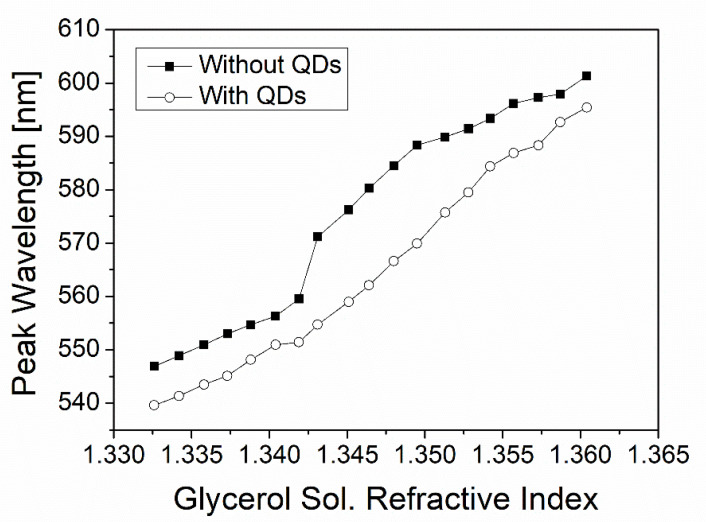
A plasmonic peak versus a local refractive index (glycerol solution index) without or with QDs adsorbed to the plasmonic metal surface. QDs’ adsorption pulls down the peak wavelengths that the y-axis represents. The x-axis denotes the glycerol solution refractive index measured by an Abbe refractometer at all concentrations.

**Figure 6 nanomaterials-12-02076-f006:**
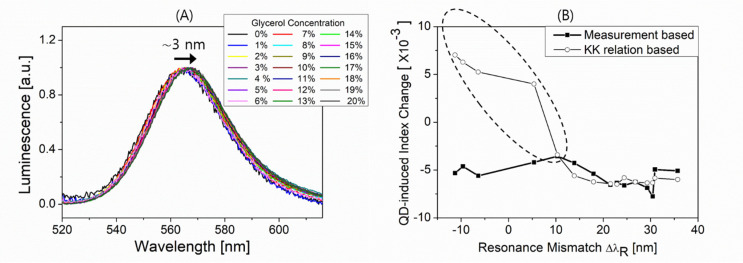
(**A**) Fluorescence of QDs immobilized on the bimetallic layer under excitation of 470 nm LED. The peak wavelength shifts from about 564 nm to 567 nm with varying surrounding glycerol concentrations from 0 to 20%. (**B**) QD-induced index change versus resonance mismatch $\Delta \lambda_{R}$. Here $\Delta \lambda_{R}=\lambda_{\mathrm{peak}}^{\mathrm{SPR}}-\lambda_{\mathrm{peak}}^{\mathrm{PL}}$ increases with glycerol concentration.

## Data Availability

The data presented in this study are available on request from the corresponding author. The data are not publicly available due to privacy.

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
