# Peer review of "Quantum Dot-Induced Blue Shift of Surface Plasmon Spectroscopy"

_nanomaterials, 2022, doi:10.3390/nano12122076_

Round 1
Reviewer 1 Report
In their manuscript, the Authors report on a blue-shift of the wavelength of the surface plasmon (SP) resonance induced by the presence of quantum dots (QDs) immobilized on the metallized surface of an optical fiber in which the SP modes are generated. Although the reported results may be of interest to the community, these are somehow not well supported by the experimental results, and also the interpretation of the effect provided by the Authors seems not sufficiently solid. Therefore, there are a number of points the Authors may should address before I can recommend publication of the paper.
- In the abstract and conclusion, the Authors affirm that <> and that <>. These rather speculative and generalistic phrases do not seem to be adequately supported or argued in the main text. The Authors should discuss in more details how their findings may be of general interest for the community working on Plasmonics and what kind of applications can benefit from them.
- The interpretation provided by the Authors on the observed blue-shift of the SP resonance induced by the presence of QDs attached on the fiber surface, it is substantially attributed to a negative variation of the local refractive index induced by light absorption from the QDs in conjunction with the long life-time of their excited states. Although this interpretation is reasonable, it is not sufficiently supported by the experimental results. Such an interpretation, in fact, call for a systematic study as a function of the QD concentration immobilized on the optical fiber surface as well as a study as a function of the light power transmitted through the fiber. Both those studies are missing in the paper and even the values of QD concentration and light power used in the experiment are not given. Also, those studies can support or refute the type of mechanism proposed by the Author for the observed blueshift. In particular, a power study may also shine light on the discrepancy found in Fig. 5B between the QD-induced refractive index change calculated from PL spectra by Kramers-Kronig relations and deduced experimentally, which the Authors tentatively attribute to a gain saturation of QDs. And issue which should display itself in a power density study. Therefore, the Authors should include such kind of studies on their manuscript or comment if they consider them not necessary.
Finally, when reviewing the article, I suggest the Authors clean up the English as there are several grammatical errors, also some sentences are difficult to read and should be rephrased to make the reading easier.
Author Response
Replying Letter
Manuscript ID: nanomaterials-1750967
Type of manuscript: Article
Title: Quantum Dot-induced Blue Shift of Surface Plasmon Spectroscopy
Authors: Than Thi Nguyen, Vien Thi Tran, Heongkyu Ju
Thanks for all reviewers’ valuable comments and advices.
Following those advices and comments, we revised the manuscript by adding more number of sentences, graphs and sections (“Materials and Methods” and “Supplementary Materials”) and corrected the sentences. All revisions are made to be seen in bold faced and underlined in the revised manuscript.
Reviewer 1
In their manuscript, the Authors report on a blue-shift of the wavelength of the surface plasmon (SP) resonance induced by the presence of quantum dots (QDs) immobilized on the metallized surface of an optical fiber in which the SP modes are generated. Although the reported results may be of interest to the community, these are somehow not well supported by the experimental results, and also the interpretation of the effect provided by the Authors seems not sufficiently solid. Therefore, there are a number of points the Authors may should address before I can recommend publication of the paper.
- In the abstract and conclusion, the Authors affirm that <> and that <>. These rather speculative and generalistic phrases do not seem to be adequately supported or argued in the main text. The Authors should discuss in more details how their findings may be of general interest for the community working on Plasmonics and what kind of applications can benefit from them.
à Thanks for your valuable advice and comments. Following the reviewer’s advice, we modified the sentences towards those including more specific applications and make them less generalized. Additions and revisions have been made as seen in bold faced and underlined sentences in Abstract and Conclusions in the revised manuscript.
- The interpretation provided by the Authors on the observed blue-shift of the SP resonance induced by the presence of QDs attached on the fiber surface, it is substantially attributed to a negative variation of the local refractive index induced by light absorption from the QDs in conjunction with the long life-time of their excited states. Although this interpretation is reasonable, it is not sufficiently supported by the experimental results. Such an interpretation, in fact, call for a systematic study as a function of the QD concentration immobilized on the optical fiber surface as well as a study as a function of the light power transmitted through the fiber. Both those studies are missing in the paper and even the values of QD concentration and light power used in the experiment are not given. Also, those studies can support or refute the type of mechanism proposed by the Author for the observed blueshift. In particular, a power study may also shine light on the discrepancy found in Fig. 5B between the QD-induced refractive index change calculated from PL spectra by Kramers-Kronig relations and deduced experimentally, which the Authors tentatively attribute to a gain saturation of QDs. And issue which should display itself in a power density study. Therefore, the Authors should include such kind of studies on their manuscript or comment if they consider them not necessary.
à Thanks for your valuable comments. We agree with the idea that the power study will give us some more interesting information, too. However, the main results of QDs-coupled plasmonic spectral shift behavior did not depend much on an incident power level as we increase it until a certain threshold level (7.5 mW) which is a maximum power level beyond which QDs could be photobleached. The fiber evanescent field of high intensity and plasmon enhanced intensity of electric fields could excite quantum dots even into photobleached states if an incident power was too high. Moreover, we worried about plasmon induced heating effects that could give rise to drastic rise of temperature in the proximity region including QDs, which could damage QDs severely. We added the fiber incident power level (3.8 mW) that we used for the main results caption (figure 5) and in the Materials and Methods section. We also made a revision to include such points in the conclusion section in the revised manuscript.
Finally, when reviewing the article, I suggest the Authors clean up the English as there are several grammatical errors, also some sentences are difficult to read and should be rephrased to make the reading easier.
àWe rephrase many of the sentences which were hard to read throughout the text.
Reviewer 2
The manuscript submitted by prof. Ju et al. demonstrate experimentally the spectral blue shift of surface plasmon resonance through the resonant coupling between quantum dots (QDs) and surface plasmons, Analysis utilizes the physical causality driven optical dispersion relation, the Kramers-Kronig (KK) relation. I have a few suggestions for this article.
Thanks for your valuable comments and advice. For each question and comments, our answers and responses are given in bold faced and underlined as follows:
- 1. Has the surface morphology of quantum dots and thin films been characterized (eg SEM、TEM)?
à We additionally provided AFM and SEM images of QDs immobilized on the bimetallic surface (Au 2nm-Ag 50 nm) on the fiber core and the glass substrate as shown in Fig S1(A)-(B) and Fig. S2(A)-(C) , respectively in a newly added “supplementary materials”.
- Whether the size or content of quantum dots also affects the blue shift.
à According to the Kramers-Kronig relation that we adopted to interpret the experimental results, any factor of optical transition related parameters such as spectral change in absorption/emission or the associated transition strength can affect the imaginary part of the electronic susceptibility. This in turn cause its real part to change. Change in QDs size or their contents would surely change these optical transition related properties, thus resulting in change in properties of the refractive index (real part of the susceptibility). Therefore, the plasmonic blue shift would change in its magnitude. We added the relevant explanation in Conclusion section of the revised manuscript as seen in bold faced and underlined sentences.
- The article said that the carboxyl-functionalized quantum dots were immobilized by EDC/NHS coupling. Is there a spectrum to prove it?
à We additionally provided the FTIR results as shown in Fig. 3 with the relevant explanation in the revised manuscript. For this measurement, we separately fabricated samples, i.e., QDs immobilized on the bimetallic thin film (2nm Au-50 nm Ag) on a glass substrate, which were fabricated through the exactly the same procedure used for the QDs immobilized on the glass fiber based bimetallic films.
- The red-shift phenomenon of plasmon spectroscopyis related to the specific energy structure of electron transition and absorption. Is the blue-shift phenomenon also related to this? If so, please analyze.
à The relevant additional explanations have been added in Results and Discussion section and Conclusion section in the revised manuscript. Additions and corrections are seen in bold and underlined sentences in the revised manuscript.
Reviewer 3
The publication entitled, ‘Quantum Dot-induced Blue Shift of Surface Plasmon Spectroscopy’, by Nguyen et al. explains a blue shift of the plasmonic resonance due to coupling effects and changes in the refractive index studied by Kramer-Kronigs relation. The work is interesting; however, I have a few concerns.
Thanks for your valuable comments and advice. For each question and comments, our answers and responses are given in bold faced and underlined as follows:
- In figure 1, the absorption at 545 nm is not visible and can be considered as an artifact. However, a more pronounced absorption is visible at 350nm and 800 nm. What do they correspond to?
à Following the reviewer’s tip, we removed the 545nm mark in the graph and rewrote the relevant explanation in the text. Broad absorption spectrum of QDs in contrast to narrow emission band is QDs-characteristic due to their electronic energy structure and carrier relaxation processes such as carrier-carrier scattering or carrier-phonon scattering. Additions and corrections are seen in bold and underlined sentences in the “Results and Discussion section” of the revised manuscript.
- Figure 3 is not properly explained. In fact, there are always two main peaks at 550 nm and 600 nm. What are they? There is a change in intensity of these peaks as the glycerol concentration changes.
à We added a number of sentences to give clearer explanations below the Fig. 5 (Fig. 3à Fig. 5) in the revised manuscript. We only consider primary peaks for comparison between cases with and without QDs as the secondary peaks disappear in cases with QDs.
(3) There is no materials and methods section. How were all the experiments performed?
à We added the section “Materials and Methods” for describing fabrication of the fiber-based samples including surface chemistry to immobilize QDs and for methodology that describes how to measure and analyze the optical spectra. Additions and corrections are seen in bold and underlined sentences in the revised manuscript.
(4) How was the fiber clad? I have read reference 21, however the authors have an additional QD layer and its deposition should be explained. In fact, the entire fiber cladding should be added to Materials and methods.
à We added experimental procedures and details about how to remove fiber cladding and QD layer immobilization in the newly added section “Materials and Methods”. Additions and corrections are seen in bold and underlined sentences in the revised manuscript
(5) There should be a SEM image showing the various layers.
à We added images of AFM and SEM for QDs immobilization in the “Supplementary Materials” and added relevant explanation in the text of the revised manuscript. Additions and corrections are seen in bold and underlined sentences in the revised manuscript.

Reviewer 2 Report
The manuscript submitted by prof. Ju et al. demonstrate experimentally the spectral blue shift of surface plasmon resonance through the resonant coupling between quantum dots (QDs) and surface plasmons, Analysis utilizes the physical causality driven optical dispersion relation, the Kramers-Kronig (KK) relation. I have a few suggestions for this article.
1. Has the surface morphology of quantum dots and thin films been characterized (eg SEM、TEM)?
2. Whether the size or content of quantum dots also affects the blue shift.
3. The article said that the carboxyl-functionalized quantum dots were immobilized by EDC/NHS coupling. Is there a spectrum to prove it?
4. The red-shift phenomenon of plasmon spectroscopy is related to the specific energy structure of electron transition and absorption. Is the blue-shift phenomenon also related to this? If so, please analyze.
Author Response

(The authors gave the same response as above.)

Reviewer 3 Report
The publication entitled, ‘Quantum Dot-induced Blue Shift of Surface Plasmon Spectroscopy’, by Nguyen et al. explains a blue shift of the plasmonic resonance due to coupling effects and changes in the refractive index studied by Kramer-Kronigs relation. The work is interesting; however, I have a few concerns.
(1) In figure 1, the absorption at 545 nm is not visible and can be considered as an artifact. However, a more pronounced absorption is visible at 350nm and 800 nm. What do they correspond to?
(2) Figure 3 is not properly explained. In fact, there are always two main peaks at 550 nm and 600 nm. What are they? There is a change in intensity of these peaks as the glycerol concentration changes.
(3) There is no materials and methods section. How were all the experiments performed?
(4) How was the fiber clad? I have read reference 21, however the authors have an additional QD layer and its deposition should be explained. In fact, the entire fiber cladding should be added to Materials and methods.
(5) There should be a SEM image showing the various layers.
Author Response

(The authors gave the same response as above.)

Round 2
Reviewer 1 Report
In the revised version of the manuscript, the Authors have answered all my doubts and modify the paper in a satisfactory way. The current version of the manuscript hits now the level to be publishable.
Reviewer 3 Report
Authors have carried out the necessary corrections. The paper can be accepted for publication.